# PLAN DEEPLY OR ESTIMATE PRECISELY?: A RESOURCE-AWARE ALPHAZERO WITH DYNAMIC QUANTILE ALLOCATION

## ABSTRACT

AlphaZero integrates deep reinforcement learning (RL) with Monte Carlo Tree Search (MCTS) and has demonstrated remarkable performance in combinatorial games. MCTS enables deep planning by leveraging learned value estimates, but in vast state spaces, these estimates require extensive sampling and often exhibit high uncertainty. While this can be mitigated with massive computational resources, such an approach is often impractical and presents two key challenges: a need for greater computational efficiency to achieve strong performance under realistic constraints, and the tendency for resource-constrained agents to develop strategies that deviate from human heuristics. In this work, we address these twin challenges by incorporating distributional RL into MCTS, replacing the scalar value estimate with a probability distribution via quantile regression. During its search, our agent dynamically increases the number of quantiles until the "action gap"—the difference between the best and second-best action values—exceeds a predefined confidence threshold. This mechanism enables the agent to autonomously trade off between deeper planning and lower value-estimation uncertainty within a fixed computational budget. We evaluated our proposed model on Four-in-a-Row—a game whose intermediate-sized state space is large enough to expose efficiency gains yet small enough to measure them precisely—and compared it with several AlphaZero variants. The model achieved higher performance while consuming fewer resources and developed effective policies with greater sample efficiency. Moreover, the model's behavioral patterns more closely resembled human heuristics compared to the other AlphaZero variants, suggesting that *how* an agent allocates its cognitive budget is crucial for emulating human-like heuristics.

## 1 INTRODUCTION

Decision-making and planning in uncertain environments are inherently challenging due to computational resource limitations and corresponding incompleteness of the gathered information. In reality, exploring all possibilities to their fullest extent is often not only inefficient but practically impossible. AlphaZero (Silver et al., 2017) stands as a prominent example. By integrating deep RL with Monte Carlo Tree Search (MCTS) (Coulom, 2006), AlphaZero has achieved superhuman performance in various combinatorial games including Go, Chess, and Shogi. The system employs policy gradient algorithms (Konda & Tsitsiklis, 1999) while using MCTS for planning on promising moves (i.e. simulating future moves). While this leads to more efficient searches, the number of state visits required by AlphaZero remains enormous. It results in the waste of computational resources, as the model requires unnecessarily deep searches even for simple problems, not only during the training process but also in real-time decision-making.

In contrast, the human, the most well-known intellectual, learns in a resource-rational manner (Callaway et al., 2022). The concept of resource-rationality means that human planning is not simply suboptimal compared to an AI like AlphaZero, but rather rational when considering the given cognitive resources, computational costs, and the value of computation. Indeed, previous studies have shown that human planning depth averages around six, whereas AlphaZero's is deeper (Zheng et al., 2022; van Opheusden et al., 2023). Direct evidence for resource-rationality can be found in comparisons between humans and AlphaZero at similar Elo ratings, where humans had fewer searches.

Human resource-rationality depends on the degree of abstraction. In comparison of expert and novice chess players, Adriaan de Groot showed that experts achieve higher performance despite having similar search depths and numbers of candidate moves (De Groot, 2008). This is attributed to their superior search direction and the higher quality of their candidate moves. This suggests that the ability to effectively utilize cognitive resources on essential tasks is linked to abstraction. Ultimately, the core of resource-rationality is to effectively manage the uncertainty of the search space by focusing on promising ones.

In this study, we propose a method to make AlphaZero more efficient—in other words, more resource-rational (Figure 1). A key challenge in this is how to handle uncertainty to determine when further exploration is needed, versus when a decision can be made confidently with the information already gathered. Given the context, distributional RL (Bellemare et al., 2017) is the one that can handle uncertainty in valuation. Unlike traditional value estimation using deep neural networks that predict a single scalar value (Mnih et al., 2015), distributional RL estimates value as a distribution using quantile regression (Dabney et al., 2018), capturing both uncertainty either inherent or epistemic.

Specifically, we set the computational resource constraint and let the AlphaZero aware of remaining budget, so that to dynamically allocate resources on either number of quantiles or tree searching depth. This provides richer information for planning—how uncertain the value estimation is, and controls the abstraction level by adjusting the number of quantiles for either coarse or fine-grained ones. It also correspondingly enables the model to explicitly determine when to terminate the forward searches, allowing the model to allocate resources where they are needed. Simply, the model is able to be more precise as the uncertainty (or variance error in Figure 1) reduces as using more quantiles to estimate value distribution. Likewise, deeper planning reduce bias-error (as in Figure 1, makes the estimation of win or lose more accurate.

In this paper, we demonstrate how the proposing resource-aware AlphaZero outperforms the conventional AlphaZero with the same computational budget. The model learns optimal moves more earlier on during training as it allocates resources more efficiently, and effectively. Moreover, the proposed model replicates human heuristics, suggesting that the model may also function in a resource-rational way.

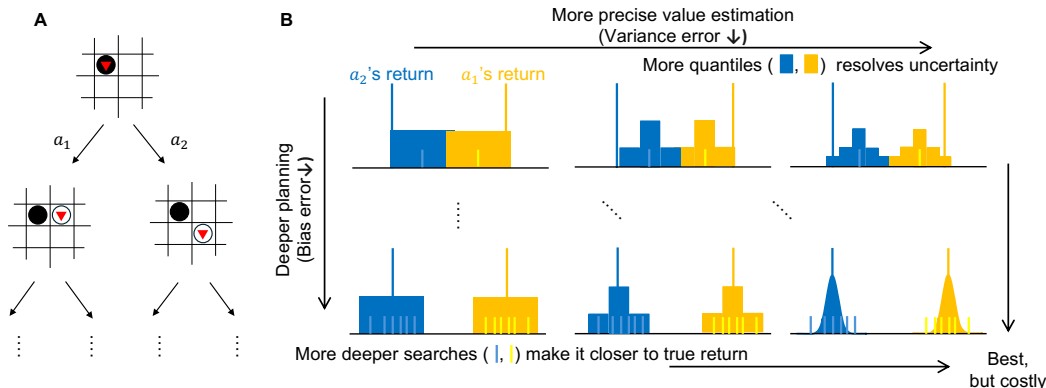

Figure 1: **Schematic illustration of the proposed method** (A) An agent (white) has two options, $a_1$ and $a_2$. (B) The proposed method tries to make less uncertain value estimation versus deeper planning. Return distributions for two actions are shown in colors ($a_1$ in yellow, $a_2$ in blue). As number of quantiles increases, the value estimation becomes less uncertain (more colored boxes with smaller width, which represents value estimation uncertainty). Also, as planning gets deeper, the searches (shown in light colored ticks) make the estimate closer to the true return. More quantiles, and deeper planning would be the best, but it is either costly or unrealistic (bottom-right corner).

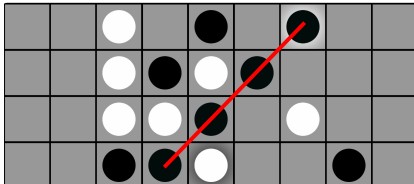

Figure 2: **Four-in-a-row task** An illustration of a board position in the four-in-a-row game. In the figure, black wins as it connects four pieces in a row (represented with a red line).

## 2 RELATED WORK

### 2.1 COGNITIVE PERSPECTIVES ON HUMAN PLANNING

During decision-making, people make plans that are mental simulations of actions and their consequences multiple steps into the future (van Opheusden et al., 2023). Such planning has an egocentric characteristic. Construal Level Theory (CLT) explains this by positing that the level of abstraction in human interpretation changes with psychological distance (Trope & Liberman, 2010). For the very next move, humans can make clear predictions, however, for several steps into the future, they can only make uncertain guesses. It suggests that human planning is also performed in a situation of low uncertainty for the foreseeable future but high uncertainty for the distant future.

The quality of planning depends on the level of abstraction. According to Adriaan de Groot's pioneering work, the difference between expert and novice chess players was not in the number of moves they searched (De Groot, 2008). Building on this, William Chase and Herbert Simon experimentally confirmed with the concept of chunking that what experts search for are patterns, not individual moves (Chase & Simon, 1973). This implies that an expert's better planning depends on how well they can abstract information into meaningful units.

It also means that in human cognition, a good plan does not simply mean an optimal one. Callaway et al. argued that humans are resource-rational, meaning that our choices are rational when considering our given cognitive resources (Callaway et al., 2022). This suggests that humans make rational choices relative to the cognitive budget. It explains why experts wins against novices in chess; the quality of their planning differs based on their ability to abstract. Their better usage of abstraction allows for a more efficient use of their resources. This resource-rational planning is empirically observed in tasks comparing humans and AlphaZero, such as the Four-in-a-row game (van Opheusden et al., 2023; Zheng et al., 2022). In a comparison with AlphaZero at a similar Elo rating, the human's planning depth was shorter, demonstrating the ability to perform good planning at a comparable level with less searching.

### 2.2 FOUR-IN-A-ROW TASK

Four-in-a-row is a two-player game, each place a black or a white piece on the board in their turn (Figure 2). The goal is to connect four pieces in a row as the name suggests. The board size is set to $4 \times 9$, although it could be any. This is because the $4 \times 9$ board is the smallest possible board size to test the optimality of the RL algorithm, since the size is proven to be where the optimal agent would not lose (Uiterwijk, 2019). The task has an intermediate state-space complexity, compared to the ones with too large state-space complexity (Go, Chess, etc.), or the ones with too small state-space complexity (Tic-tac-toe) (van Opheusden et al., 2023). The state-space complexity (Zhang, 1999) quantifies the number of possible states, acts as an estimate of the difficulty of the games.

The intermediate state-space complexity of four-in-a-row is especially beneficial for our study. It is large enough to show benefits of the proposed model's efficiency, yet small enough that both humans and RL agents can visit a plausible number of states during planning. This keeps the computational budget realistic and tractable, and allows more contrastive comparison of other AlphaZeros (or humans) to the proposed method. For example, in a game with large state-space complexity such as Go, AlphaZero may require an unrealistically enormous number of searches, and make the comparison intractable. Moreover, the intermediate state-space allows us to replace the state value estimation to action value estimation scheme, makes each node search in MCTS identical to a deeper search tree.

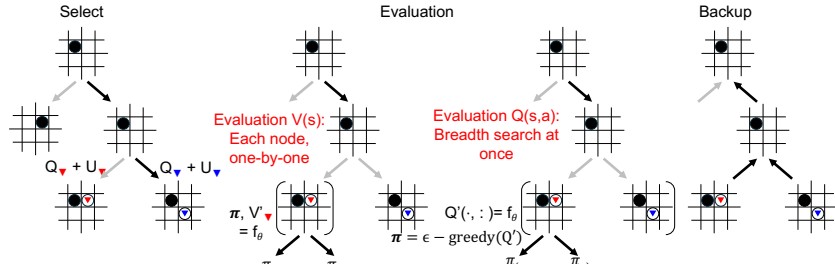

Figure 3: **AlphaZero's MCTS procedures.** It begins with the given state $s$ (top row) and "selects" and "expands" a node (represented in black arrow) based on the sum of values ($V$ or $Q$) and $U$ (exploration term in UCT). During the "evaluation" stage, AlphaZero's value network estimates state value $V(s)$ or action value $Q(s, a)$. Then, it performs backups, back-propagating the simulation results through the parent nodes in the tree. Then, it iterates all over from "select" until it terminates or meets the maximum number of searches ($N_{\text{MCTS}}$).

## 2.3 ALPHAZERO

AlphaZero (Silver et al., 2017) is an algorithm that integrates deep RL with MCTS to discover optimal policies in environments with a large state-space. MCTS consists of four stages: selection, expansion, evaluation, and backpropagation (Figure 3). Note that originally the evaluation includes rollouts (simulations), which are replaced with value estimates using the value network in AlphaZero. In the selection stage, strategies like the Upper Confidence Bound for Trees (UCT) (Kocsis & Szepesvári, 2006) are used to balance exploration and exploitation by selecting the most promising node. Although there is no existing proof for convergence of AlphaZero, the convergence characteristics of UCT is provided (Kocsis & Szepesvári, 2006) (refer to Appendix B.2.2)

It is also worth noting that MCTS, particularly as implemented in AlphaZero, shares key behavioral characteristics with Best-First Search (BFS) (Pearl, 1984). In BFS, the search proceeds by always expanding the most promising node based on a complete evaluation of all candidate nodes. Similarly, in MCTS, the selection step guided by UCT prioritizes nodes that exhibit either high estimated value or low visitation count, effectively treating those nodes as "promising" under a utility function, which serves as a proxy for full evaluations.

## 2.4 QUANTILE REGRESSION FOR VALUE DISTRIBUTION ESTIMATION

QR-DQN (Dabney et al., 2018) is a distributional RL algorithm designed to learn the value distribution of a given state-action pair by quantile regression. This method enables the agent to effectively capture uncertainty in returns while providing a more detailed representation of the value distribution compared to non-distributional RL algorithms, such as DQN (Mnih et al., 2015). QR-DQN employs equally distributed quantiles to represent value distributions, which is proven to be the minimizer of Wasserstein distance between the true return distribution and the quantile regression-based value distribution estimation. Specifically, QR-DQN minimizes the following Huber loss:

$$\mathcal{L} = \frac{1}{N} \sum_{i=1}^{N} \sum_{j=1}^{N} \rho_\kappa^{\tau_i} \left( y_j - \hat{Z}^{(i)}(s, a) \right) \tag{1}$$

where $\hat{Z}^{(i)}(s, a)$ denotes the $i$-th predicted quantile and $y_j$ is the $j$-th target quantile obtained via the Bellman update. Here, $\rho_\kappa^{\tau_i}(\cdot)$ is the quantile Huber loss function, designed to combine the robustness of absolute error with the smoothness of squared error.

## 3 RESOURCE-AWARE ALPHAZERO

In this work, we propose resource-aware AlphaZero, an AlphaZero that dynamically allocates computational resources to either deepening planning in MCTS or reducing uncertainty in value estimation. To do so, we adapted MCTS to incrementally increase the number of quantiles $N_\tau$ until the

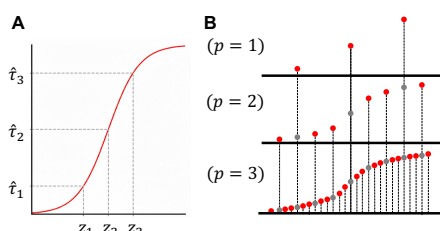

Figure 4: **Quantile-adaptive value distribution estimation** (A) Quantile regression. Value distribution and corresponding quantile value ($Z_1, Z_2, Z_3$) for $N_\tau = 3$ are aligned. (B) Quantile-adaptive evaluation. New quantiles are added while maintaining existing quantiles as $p$ increases.

model is sufficiently confident that the best action is actually the best. Then, we correspondingly modified the entire planning process to share computational resource budget, for both number of quantiles $N_\tau$ and number of node searches $N_{\text{MCTS}}$.

### 3.1 RESOURCE-AWARE PLANNING

---

**Algorithm 1:** `Resource-aware MCTS (with QR-DQN)`

**Input:** state $s$, computational resource $\mathcal{R}$
**Output:** policy $\pi(s, :)$
Initialize root node with state $s$
**while** $\mathcal{R} > 0$ **do**
    node $\leftarrow$ `NodeSelect`$(s)$;
    $q, \mathcal{R}' \leftarrow$ `Quantile-adaptive Evaluation`$(\text{node}, \mathcal{R})$;
    `Backup`$(q)$;
    $\mathcal{R} \leftarrow \mathcal{R}' - \Delta_{\text{Depth}}$
**return** $\pi(s, :) \leftarrow \epsilon\text{-}greedy(q)$

---

First, we modified the MCTS, aiming for the agent to plan under a computational resource constraint, as opposed to the fixed number of searches ($N_{\text{MCTS}}$) in AlphaZero's conventional MCTS. Specifically, it starts with a predefined resource budget, $\mathcal{R}$. The process is exactly the same as the original MCTS except it continues until the computational resource $\mathcal{R}$ runs out through `Resource-aware MCTS` (Algorithm 1):

$$\pi(s, :) = \text{Resource-aware MCTS}(s, \mathcal{R})$$

where $\pi(s, :)$ is policy in the given state $s$, and $\mathcal{R}$ is the computational resource constraint that would be discounted every MCTS step.

There are two kinds of discounting of resources: (1) depth-wise discount for an increased planning depth; it discounts a fixed amount of resource ($\Delta_{\text{Depth}}$); (2) quantile-adaptive discount in `Quantile-adpative Evalaution`, where also a certain amount of resource is discounted as the number of quantiles $N_\tau$ increases.

### 3.2 QUANTILE-ADAPTIVE EVALUATION OF VALUE DISTRIBUTION

For designing resource-aware AlphaZero, we devised a novel approach for the quantile regression for estimating value distribution (Figure 4). We developed this in a quantile-adaptive manner, to make the agent handle value estimation uncertainty autonomously. To do so, we parameterized $N_\tau$ in power law as $N_\tau = 3^p$, where $p$ is the exponent for determining $N_\tau$. It allows the previously sampled quantile values to be used even as $p$ (i.e., $N_\tau$) increases (Figure 4B).

The algorithm starts with $p = 1$, and computes value distribution with quantile regression as QR-DQN does (Algorithm 2). The while loop terminates when the action gap (i.e.,) is greater than the predefined threshold $\theta = 0.1$, or until $p = 4$. We further explained the implementation of the network, and training procedures in great detail in Appendix C.

Taken together, the proposed resource-aware AlphaZero models possess two key attributes: (1) a variable number of quantiles, and (2) MCTS with quantile regression-based value estimation. These give freedom the model dynamically allocate resources as needed. We provided proof sketch of convergence of the model as well (Appendix B).

---

**Algorithm 2:** `Quantile-adaptive Evaluation`

---

**Input:** state $s$, computational resource $\mathcal{R}$
**Output:** remaining resource $\mathcal{R}'$
Initialize $Q_{best}, Q_{second\ best}$ with 0, and $p = 1$
**while** $(Q_{best} - Q_{second\ best}) \leq \theta$ *or* $p < 5$ **do**
    $z \leftarrow$ `Quantile-regression`$(s, 3^p)$
    $q' \leftarrow \mathbb{E}(z)$
    $Q_{best}, Q_{second\ best} \leftarrow$ `sort`$(q', \text{descending})([0, 1])$
    $p \leftarrow p + 1$
    $\mathcal{R}' \leftarrow \mathcal{R} - \Delta_{\text{quantile}}$
**return** $q'$, $\mathcal{R}'$

---

### 3.3 QUANTILE-ADAPTIVE RL BACKBONES

For the value estimation, we used modified distributional RL backbones. These, EQR-QAC and EQR-DQN, are the AlphaZero variants using resource-aware MCTS ("E" here stands for efficient). The computational resource budgets for these models were matched to those of their non-resource-aware counterparts (i.e. QR-QAC and QR-DQN), where defined by $N_{\text{MCTS}}$ and $N_\tau$:

$$\mathcal{R} = f(N_\tau) \times \Delta_{\text{Quantile}} + g(N_{\text{MCTS}}) \times \Delta_{\text{Depth}} \tag{2}$$

where $f$ and $g$ are non-linear mappings from $N_\tau$ and $N_{\text{MCTS}}$ to the actual resource consumption. In practice, we set $f(N_\tau) = \log_3 N_\tau$ and $g(N_{\text{MCTS}}) = N_{\text{MCTS}}$ (i.e., the identity function), which simplifies the implementation (Refer to Appendix C.3.2 for ablation study of different non-linear function). There are a full list of the model in Appendix A.2, and implementation details in Appendix C.

## 4 RESULTS

### 4.1 PERFORMANCE COMPARISON

We first compared the performance of models in terms of Elo rating (Elo, 1978) that was measured through a round-robin tournament among the trained models. It increased following a win and decreased following a loss. We compared $12 \times 5 + 2 \times 5 = 70$ models [1] in total (refer to Appendix D.1 and D.2 for more details).

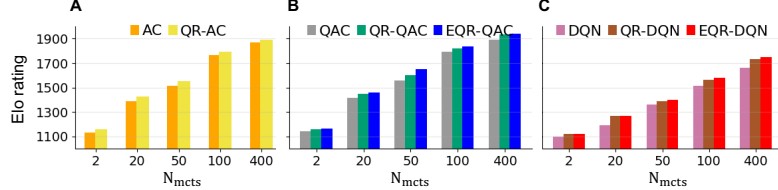

Figure 5: **Elo ratings of models** Elo ratings of agents are grouped to (A) AC models with state-value network, (B) AC-based models with action-value network, (C) DQN-based models with action-value network. Here, all models are in case of $N_{\text{MCTS}} = 400$ and $N_\tau = 3^4$.

Here, we found several consistent patterns in model performance (Figure 5). First, models based on the AC architecture outperformed their DQN-based counterparts, reflecting the advantage of AlphaZero-style learning through direct policy updates via MCTS. Second, among the AC-based variants, those employing the action-value approximation (QAC, QR-QAC, EQR-QAC) surpassed those using state-value estimation (AC, QR-AC). Third, distributional RL (i.e. quantile regression) led to performance improvement with higher Elo rate as $N_\tau$ increases. Lastly, models with resource-aware MCTS (EQR-QAC and EQR-DQN) outperformed others.

---

[1]**12**: AC, QAC, QR-AC, four QR-QAC with different $N_\tau$s, DQN, and four QR-DQN with different $N_\tau$s; **5**: five different $N_{\text{MCTS}}$; **2**: EQR-QAC and EQR-DQN.

These results may be partially explained by the structural characteristics of the models. The action-value approximation may outperform in the current $4 \times 9$ board; however, it is unlikely to do so in larger board sizes. Distributional RL models seem to generally help to model value uncertainty. Most importantly, resource-aware models (EQR-QAC and EQR-DQN) that dynamically adjust the planning depth and $N_\tau$ showed stark improvements over their fixed-resource counterparts, highlighting the benefit of adaptive computation in planning. Conceptually, planning depth is akin to how many moves ahead a chess player thinks. A novice might only consider their next immediate move (a shallow depth), while a grandmaster visualizes complex sequences of moves and counter-moves far into the future (a deep depth).

## 4.2 COMPARISON OF LEARNING EFFICIENCY

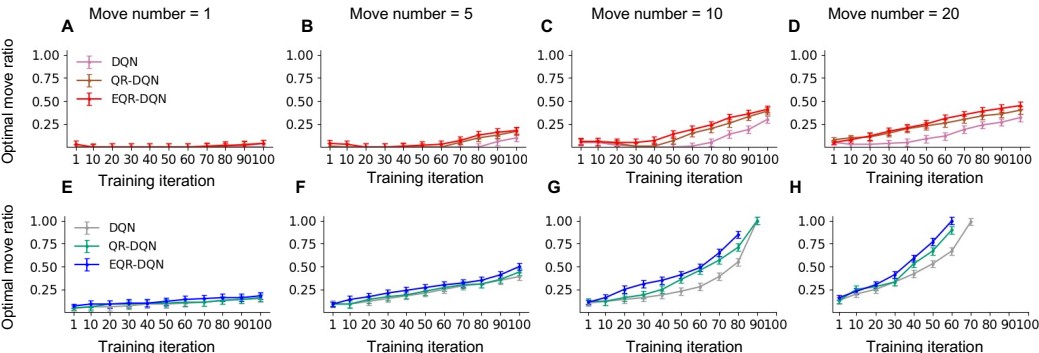

Figure 6: **Learning Efficiency of Resource-Aware Models and Counterparts** Optimal move ratio is plotted as a function of training iterations. (A-D) EQR-QAC's results for the first, 5th, 10th, and 20th move. (E-H) EQR-DQN's results for the first, 5th, 10th, and 20th move. Here, the optimal move is defined by pure MCTS search with $N_{\text{MCTS}} = \infty$, meaning that the fully searching model.

We tested if the resource-aware model learns more efficiently from the beginning of the training phase. To do so, resource-aware models are compared to their counterparts. For example, EQR-QAC has compared to QR-QAC and QAC; and EQR-DQN has compared to QR-DQN and DQN. We analyzed the optimal move ratio as a function of the move number (Figure 6), which defined by the sequential order of a move in a game.

We found that the resource-aware model learns optimal moves early across all move numbers. This pattern remained consistent in the same computational budgets (i.e., $N_{\text{MCTS}}$) were applied, although a larger budget naturally led to faster learning for both models (Appendix **??**).

## 4.3 INTERPLAY BETWEEN PLANNING DEPTH AND VALUE ESTIMATION UNCERTAINTY

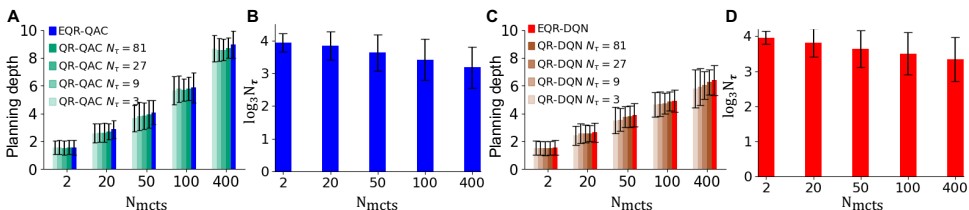

Figure 7: **Interplay between planning depth and value estimation uncertainty** (A) Planning depth of QAC models. (B) $N_\tau$ of QAC models in logarithmic scale. (C) Planning depth of DQN models. (D) $N_\tau$ of DQN models in logarithmic scale. Each bar shows the average computed from matches between resource-aware models and their corresponding non-resource-aware counterparts. Error bars indicate standard deviations.

Although resource-aware AlphaZero outperformed the others, it is unclear if the computational resources are effectively managed during inference. To test this, we compared resource-aware Al-

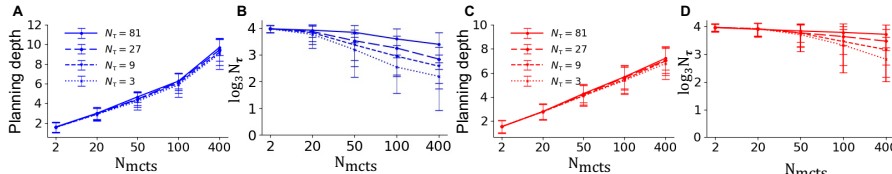

Figure 8: **Resource allocation profile of resource-aware AlphaZero against non-resource-aware ones** (A) Planning depth of EQR-QAC models. (B) $N_\tau$ of EQR-QAC models in logarithmic scale. (C) Planning depth of EQR-DQN models. (D) $N_\tau$ of EQR-DQN models in logarithmic scale. The lines the counterpart opponent (QR-QAC for EQR-QAC and QR-DQN for EQR-DQN) with different budget ($N_\tau$), not represent the budget of the resource-aware MCTS.

phaZero models to their counterparts, measuring planning depth and $N_\tau$ during MCTS planning (Figure 7). Here, planning depth is defined by the maximum search depth visited during MCTS. Note that although resource-aware models has no $N_{\text{MCTS}}$, but grouped together with the other models where their $\mathcal{R}$ is calculated with the same $N_{\text{MCTS}}$ (equation 3). The results showed that resource-aware AlphaZero prioritizes deeper planning, while allocating fewer resources to $N_\tau$ as $N_{\text{MCTS}}$ (i.e., computational budget) increases (Figure 7). It shows that the model adaptively allocates its resource during inference.

Then, we further analyzed how resource-aware AlphaZero behaves as a function of an opponent's computational resource budgets (Figure 8). Specifically, we analyzed matches between resource-aware AlphaZeros and their counterpart models with different budgets. For instance, EQR-QAC's planning depth and $N_\tau$ are measured when it is playing against QR-DQN with $N_\tau = \{3, 9, 27, 81\}$. This allow us to analyze how the resource-aware AlphaZero adaptively changes its resource allocation properties. While all of them follows a general trend of increasing planning depth with larger $N_{\text{MCTS}}$ (i.e., more budget), the degree of increase was less when the opponent has smaller $N_\tau$ (Figure 8A and C). Similarly, although both models shared the trend of decreasing $N_\tau$ as $N_{\text{MCTS}}$ increased, this reduction was more substantial when facing opponents with smaller $N_\tau$ (Figure 8B and D). It indicates that the resource-aware AlphaZero is adaptive to opponents, in a way that uses resources as needed.

We also tested this resource allocation profile in a game as training progresses (Appendix D.4). Regardless of training iterations, model has the same profile, which is prioritizing planning depth while using less number of quantiles.

### 4.4 COMPARISON TO HUMAN BEHAVIOR

Finally, we examined how closely the resource allocation behavior of our agent resembles that of humans (Figure 9). To this end, we compared the model's behavior with known human heuristics in four-in-a-row, particularly those related to valuation, such as distance to the board center (Figure 9A and E), distance to own pieces (Figure 9B and F), number of threats made (Figure 9C and G), and number of threats defended (Figure 9D and H) (Kuperwajs et al., 2022; van Opheusden et al., 2023). Following prior work, we defined threats as proactive attempts to build winning conditions (e.g., creating three-in-a-row and two-in-a-row with no opponent's pieces around), and defenses as actions that block the opponent's threats.

We found that the resource-aware AlphaZero is the best model that replicates human heuristics (Figure 9). Especially, EQR-QAC models with an $N_{\text{MCTS}}$ between 50-100 consistently showed the closest match with any heuristics. All the others including non-resource-aware models showed that the variety of computational budgets does not capture human heuristics (Appendix D). For example, EQR-DQN failed to achieve a proper fit even with the maximum budget ($N_{\text{MCTS}}=400$) for "number of threats made" and "number of threats defended".

Lastly, to test the speculation from CLT that planning becomes more abstract with greater psychological distance, we analyzed how the number of quantiles changes as the search depth increases within a single MCTS search (i.e., a single planning instance) for each move number (Figure 10).

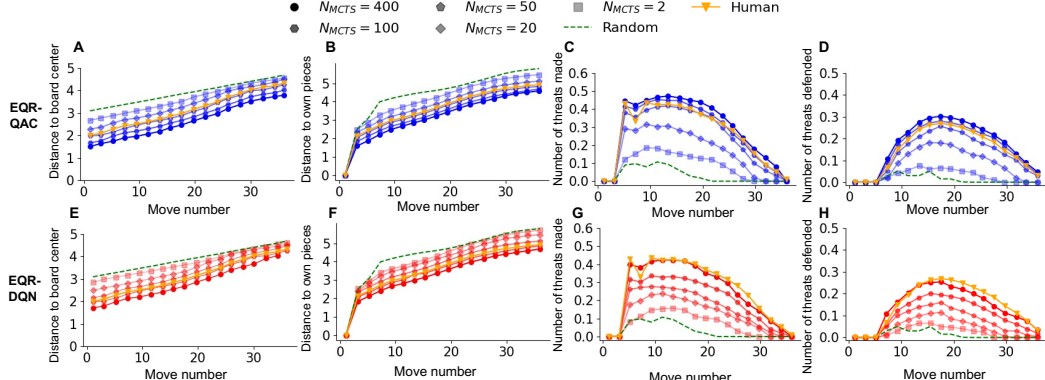

Figure 9: **Comparison to human behavior** We evaluated how closely QA-MCTS-based models resemble human play behavior under different $N_{\text{MCTS}}$ settings. (A–D) show the performance of EQR-QAC, while (E–H) correspond to EQR-DQN. Each model is compared against human behavioral data (red) and a random policy baseline (green), which selects actions uniformly at random. The subplots represent metrics related to spatial positioning and threat-related behavior.

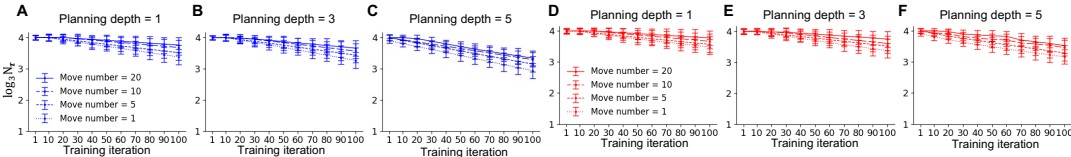

Figure 10: **Changes in the number of quantiles as a function of search depth** In a single planning instance, the number of quantile is measured for some move numbers as a function of search depth. (A-C) EQR-QAC's and (D-F) EQR-DQN's. Error bars represent standard deviations.

Indeed, the model used smaller number of quantiles as planning gets deeper regardless of move number.

## 5 CONCLUSION

In this study, inspired by cognitive perspective of human planning, we proposed a resource-aware MCTS that adaptively modify number of quantiles. By doing so, the proposed model could learn and behave efficiently under computational resource budget, in a resource-rational manner.

Specifically, we introduced resource-aware MCTS to AlphaZero with various RL models such as AC and DQN. We tested a total of 70 models, among which the proposed EQR-QAC achieved the highest Elo rating, demonstrating the effectiveness of our approach. Notably, the resource-aware AlphaZero (EQR-QAC and EQR-DQN) dynamically adjusted its planning depth based on the characteristics of the opponent, and in certain settings, exhibited behavior patterns that resembled human planning heuristics.

Although the current implementation of resource-aware MCTS relies on action gap thresholds to allocate resources, the experimental results showed that these heuristics functioned as strong inductive biases. Resource-aware AlphaZero consistently balanced between deeper planning and reducing value estimation uncertainty across diverse opponents and resource conditions. This suggests promising potential for extending the framework with learned resource allocation mechanisms in future work.

Furthermore, more works should be done focusing on developing it to go beyond the limited domain of four-in-a-row, applying the proposed framework to broader settings—such as robotics control, and real-time decision-making would be an important step toward validating the practicality and generality of resource-aware AlphaZero.

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

APPENDIX

## CODE AVAILABILITY

The code will be made publicly available upon publication.

## A  RESOURCE-AWARE PLANNING

First, we modified the MCTS, aiming for the agent to plan forward under a computational resource constraint, as opposed to the fixed number of searches ($N_{MCTS}$) in AlphaZero's conventional MCTS. Specifically, it starts with a predefined resource budget, $\mathcal{R}$ (Algorithm 1).

In here, the process is exactly the same as the conventional MCTS of AlphaZero except it continues until the computational resource $\mathcal{R}$ runs out. Through the modified MCTS (`Resource-aware MCTS`; Algorithm 1), the agent plans by simulating future moves, and gets the probability of actions in the given state $s$, policy $\pi$:

$$\pi(s, :) = \text{Resource-aware MCTS}(s, \mathcal{R})$$

where $\mathcal{R}$ is the computational resource constraint that would be discounted every MCTS step. Note that in the algorithm we showed that policy calculated using $\epsilon$-greedy of action value ($q$) assuming QR-DQN has used for quantile regression, however, this can be replaced with policy network as the original AlphaZero does.

Note that there are two kinds of resource discounting: (1) depth-wise discount for each node expansion it discounts a fixed amount of resource ($\Delta_{Depth}$) for an increased planning depth; (2) quantile-adaptive discount in `Quantile-adpative Evalaution`, where also a certain amount of resource is discounted after increasing the number of quantiles $N_\tau$.

### A.1  QUANTILE-ADAPTIVE EVALUATION OF VALUE DISTRIBUTION

Quantile regression is a method to learn the value distribution by regression of each quantile to the returns in the cumulative density function (Figure 4A). We developed this in a quantile-adaptive manner, to make the agent handle value estimation uncertainty autonomously. To do so, we parameterized $N_\tau$ in power law as $N_\tau = 3^p$, where $p$ is the exponent for determining $N_\tau$. Here, although any base can work in theory, we specifically used 3 as the base because the unique minimizer of quantile regression loss (Dabney et al., 2018) can be re-used even with the increased $N_\tau$ (Figure 4B).

The algorithm starts with $p = 1$, and computes value distribution with quantile regression as QR-DQN does (Algorithm 2). The while loop terminates when the action gap is greater than the predefined threshold $\theta = 0.1$, or until $p = 4$. We set the maximum $N_\tau$ to $3^4$, because it was enough to train QR-DQN (which has no quantile-adaptive feature) in the four-in-a-row game. `Quantile-regression` is the value estimation network as in QR-DQN, but not exactly the same since we also used quantile regression for the state-value distribution, or advantage distribution depending on the model. We further explained the implementation of the network, and training procedures in great detail in Appendix C.

There are some things worth to note. First, the thresholding condition can be smoothed with non-linear function that maps to probability to halt increasing $N_\tau$ (such as $P(\sigma(w \cdot (Q_{best} - Q_{second\ best}) + b))$, where $w$ and $b$ are weight and bias), however, it also introduced instability of performance during training in practice. Second, although the algorithm seems to run `Quantile-regression` in every iteration, practically it runs partially for the newly added quantiles.

Taken together, the proposed resource-aware AlphaZero models possess two key attributes: (1) a variable number of quantiles, and (2) MCTS with quantile regression-based value estimation. We provided separate convergence proofs for each component: one demonstrating that value estimation converges to the same stationary points even when the number of quantiles varies, and another proving the convergence of MCTS with quantile regression-based value estimation (Refer to Appendix B.1). Note that these proofs were conducted independently, and additional work is required to establish convergence for the full algorithm in a joint manner.

## A.2 Variants of RL backbones

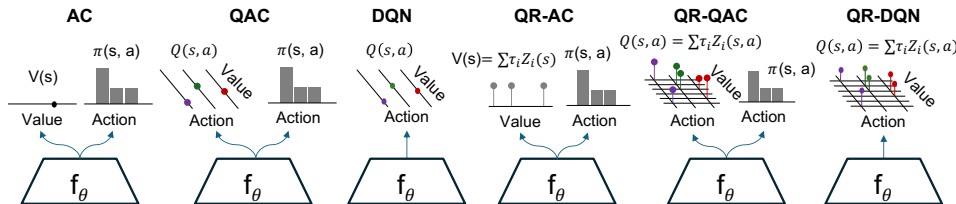

Figure 11: **A schematic view of models** The figure compares six network architectures for reinforcement learning. AC estimates the state-value function $V(s)$, while QAC and DQN estimate the action-value function $Q(s, a)$. QR-AC extends AC by approximating the distribution of $V(s)$ through quantile regression. Similarly, QR-QAC and QR-DQN extend QAC and DQN by modeling the distribution of $Q(s, a)$ using multiple quantile outputs.

To validate the effectiveness of resource-aware planning in AlphaZero, we first categorized the possible advances of our approach: (1) Action-value based approach makes MCTS process quicker as it evaluates one breadth at once (all possible actions in the given state $s$); (2) Distributional value estimation with quantile regression plays a key role for handling uncertainty during planning; (3) resource-aware MCTS is better in terms of both performance and efficiency, especially using fewer computations.

To make the experiment fair in assessing which advancement contributed the most, we designed AlphaZero variants with various types of RL backbones and various resource constraints (Figure 11). For the RL backbones, including the one for original AlphaZero (actor-critic; AC), we have quantile-regression version (QR-AC), action-value version (QAC), and quantile-regression action-value version (QR-QAC). For QR-QAC, we had four variants with $N_\tau \in \{3, 9, 27, 81\}$. Note that for the quantile regression-based state value network, we modified equation 1 to calculate $\hat{Z}$ as a function of state.

We also used DQN as RL backbones to compare with AC-based variants under the same planning framework to isolate the effect of policy gradient. Likewise, we have the original DQN and quantile-regression version (QR-DQN; also four variants with $N_\tau \in \{3, 9, 27, 81\}$). Note that these DQN-based ones used $\epsilon$-greedy algorithm for policy (refer to Appendix C.3 for more details). For MCTS, we tested $N_{\text{MCTS}} = \{2, 20, 50, 100, 400\}$ following the previous literature tested AlphaZero on four-in-a-row (Zheng et al., 2022).

Lastly, we propose EQR-QAC and EQR-DQN, by appying the resource-aware MCTS to QR-QAC and QR-DQN, respectively ('E' denotes efficiency in terms of computational cost). The computational resource budgets for these models were matched to those of their non-resource-aware counterparts, with $N_{\text{MCTS}} \in \{2, 20, 50, 100, 400\}$ and $N_\tau = 81$.

The total computational resource budget was defined as follows:

$$\mathcal{R} = f(N_\tau) \times \Delta_{\text{Quantile}} + g(N_{\text{MCTS}}) \times \Delta_{\text{Depth}} \tag{3}$$

where $f$ and $g$ are non-linear mappings from $N_\tau$ and $N_{\text{MCTS}}$ to the actual resource consumption. In practice, we set $f(N_\tau) = \log_3 N_\tau$ and $g(N_{\text{MCTS}}) = N_{\text{MCTS}}$ (i.e., the identity function), which simplifies the implementation: each iteration incurs a fixed cost of $\Delta_{\text{Quantile}}$ and $\Delta_{\text{Depth}}$.

## B    Proof sketch

Here, we provide how mathematical analysis how resource-aware MCTS would converge. This proof sketch is composed of three steps: (1) Consistency of value estimation with quantile-adaptive evaluation; (2) Convergence of MCTS; (3) Proof sketch of resource-aware MCTS and quantile-adaptive evaluation (i.e., the proposed model).

### B.1 Consistency of Quantile Estimation Under Variable Number of Quantiles

We first establish the convergence properties when the number of quantiles varies. This serves as the foundation for our resource-aware MCTS.

Let $X_1, \ldots, X_n$ be i.i.d. random variables with cumulative distribution function (CDF) $F$ and probability density function (PDF) $f$. The theoretical $\tau$-quantile is defined as:

$$q_\tau := \inf\{x : F(x) \geq \tau\}. \tag{4}$$

The empirical CDF is:

$$\hat{F}_n(x) = \frac{1}{n} \sum_{i=1}^{n} \mathbf{1}\{X_i \leq x\}, \tag{5}$$

and the corresponding empirical quantile is:

$$\hat{q}_\tau := \inf\{x : \hat{F}_n(x) \geq \tau\}. \tag{6}$$

In practical implementations, we often estimate $K_n$ quantiles at positions $\tau_1, \ldots, \tau_{K_n}$ where $K_n$ may vary based on computational constraints.

We begin with the classical result:

**Lemma 1** (Bahadur Representation (Bahadur, 1966)). *If $F$ is differentiable at $q_\tau$ with $f(q_\tau) > 0$, then*

$$\hat{q}_\tau = q_\tau + \frac{\tau - \hat{F}_n(q_\tau)}{f(q_\tau)} + r_n, \quad a.s. \tag{7}$$

This representation decomposes the quantile estimation error into a leading linear term and a remainder term.

#### B.1.1 Asymptotic Properties of Empirical Quantiles

Then, we rely on the Dvoretzky-Kiefer-Wolfowitz inequality:

**Lemma 2** (DKW Inequality (Dvoretzky et al., 1956)). *For any $\varepsilon > 0$,*

$$\Pr\left(\sup_x |\hat{F}_n(x) - F(x)| > \varepsilon\right) \leq 2e^{-2n\varepsilon^2}. \tag{8}$$

We now state our main consistency result.

**Theorem 1.** *Let $X_1, \ldots, X_n$ be i.i.d. random variables with CDF $F$ and PDF $f$, satisfying:*

- *$F$ is strictly increasing and continuously differentiable in neighborhoods of each $q_{\tau_k}$;*

- *$f(q_{\tau_k}) > 0$ for all $k$.*

*If $K_n = o(n/\log n)$, then:*

$$\max_{1 \leq k \leq K_n} |\hat{q}_{\tau_k} - q_{\tau_k}| \xrightarrow{a.s.} 0. \tag{9}$$

*Moreover, if $F$ has finite moments of order $p > 1$, then:*

$$\max_{1 \leq k \leq K_n} \mathbb{E}[|\hat{q}_{\tau_k} - q_{\tau_k}|^p] \to 0. \tag{10}$$

*Proof.* From Lemma 1,

$$\hat{q}_{\tau_k} = q_{\tau_k} + \frac{\tau_k - \hat{F}_n(q_{\tau_k})}{f(q_{\tau_k})} + r_n.$$

Applying the union bound to the DKW inequality,

$$\Pr\left(\max_k |\hat{F}_n(q_{\tau_k}) - F(q_{\tau_k})| > \varepsilon\right) \leq 2K_n e^{-2n\varepsilon^2}.$$

Since $K_n = o(n/\log n)$, the series

$$\sum_n \Pr\left(\max_k |\hat{F}_n(q_{\tau_k}) - F(q_{\tau_k})| > \varepsilon\right) < \infty.$$

Let us define the events

$$A_n := \left\{\max_k |\hat{F}_n(q_{\tau_k}) - F(q_{\tau_k})| > \varepsilon\right\}.$$

By the Borel-Cantelli lemma (cf. Feller, 1968), if $\sum_n \Pr(A_n) < \infty$ and the $A_n$ are independent, then

$$\Pr(A_n \text{ i.o.}) = 0.$$

Thus, $\max_k |\hat{F}_n(q_{\tau_k}) - F(q_{\tau_k})| \to 0$ a.s.

Let $C = \min_k f(q_{\tau_k}) > 0$, then:

$$\max_k |\hat{q}_{\tau_k} - q_{\tau_k}| \le \frac{1}{C} \max_k |\hat{F}_n(q_{\tau_k}) - F(q_{\tau_k})| + o(1),$$

which converges to 0 a.s., proving the first part.

For the second part, apply the dominated convergence theorem under finite $p$-th moments. $\square$

## B.2 CONVERGENCE OF MCTS

We now review the convergence properties of MCTS, particularly focusing on the Upper Confidence Bound applied to Trees (UCT) algorithm introduced by Kocsis & Szepesvári (2006).

### B.2.1 UCT ALGORITHM FRAMEWORK

MCTS with UCT constructs a search tree incrementally through four phases:

- **Selection**: Traverse the existing tree using the UCB1 policy until reaching a leaf node.
- **Expansion**: Add one or more child nodes to the leaf node.
- **Simulation**: Run a random simulation (rollout) from the newly expanded node.
- **Backpropagation**: Update values along the path from the expanded node to the root.

The UCB1 policy selects actions based on:

$$a = \arg\max_{a'}\left[Q(s, a') + C\sqrt{\frac{\log N(s)}{N(s, a')}}\right], \tag{11}$$

where $Q(s, a')$ is the estimated value, $N(s)$ is the visit count for state $s$, $N(s, a')$ is the count for taking action $a'$ in $s$, and $C$ is an exploration constant.

### B.2.2 CONVERGENCE OF UCT (KOCSIS & SZEPESVÁRI, 2006)

As this convergence proof is rely on Kocsis & Szepesvári's work, please refer to the paper for more information. The convergence proof of UCT begin as follow:

**Theorem 2** (Kocsis & Szepesvári (2006)). *Let the environment be a finite Markov Decision Process (MDP) with bounded rewards in $[0, 1]$. Then, as the number of simulations $n \to \infty$, the value estimated by UCT converges almost surely to the optimal value:*

$$\hat{V}_n(s) \xrightarrow{a.s.} V^*(s).$$

**Proposition 1** (Xu et al. (2020)). *Using a polynomial exploration bonus instead of the standard* log-*based UCB1 improves convergence rate:*

$$\mathcal{O}\left(n^{-1/(2+\beta)}\right),$$

*where $\beta > 0$ is the exploration exponent.*

### B.3 Proof sketch of Resource-aware MCTS and Quantile-adaptive evaluation

Let $\hat{q}_{\tau_k}^{(n)}$ denote the empirical quantile estimates, and let $\hat{V}_n$ be the estimated value from MCTS using quantile-based statistics.

**Theorem 3** (Joint Convergence). *Let MCTS use quantile estimates $\{\hat{q}_{\tau_k}^{(n)}\}_{k=1}^{K_n}$ with $K_n = o(n/\log n)$. Under the conditions of Theorems 1 and 2:*

- *Every action is tried infinitely often: $N(s, a) \to \infty$ for all valid $(s, a)$.*

- *The overall regret is bounded by the sum of estimation errors from MCTS and quantile statistics.*

*Proof.* We decompose the total error into two parts:

$$|\hat{V}_n(s) - V^*(s)| \leq |\hat{V}_n(s) - \tilde{V}_n(s)| + |\tilde{V}_n(s) - V^*(s)|,$$

where $\tilde{V}_n(s)$ is the MCTS estimate assuming perfect quantile information.

- The second term $\to 0$ by UCT convergence.

- For the first term, we use the Lipschitz continuity of value functions in the quantile space:

$$|\hat{V}_n(s) - \tilde{V}_n(s)| \leq L \cdot \max_k |\hat{q}_{\tau_k}^{(n)} - q_{\tau_k}|,$$

where $L$ is the Lipschitz constant depending on the MDP structure (Bai et al., 2023).

By Theorem 1, since $K_n = o(n/\log n)$ and quantiles converge almost surely, we conclude:

$$\hat{V}_n(s) \xrightarrow{\text{a.s.}} V^*(s).$$

$\square$

One thing must be noted that this convergence proof sketch relies on previous literature, which is based on various assumptions. Through this, we can prove that it is a model that can converge when there are enough resources, but this does not mean that it also converges in resource-rational setting (i.e., where the computational resource budget is limited). This was explained with empirical evidence in the main text.

## C  Details of experimental setup

### C.1  Implementation of AlphaZero

The neural network used in this study is based on AlphaZero architecture (Silver et al., 2017). Due to the board size of $4 \times 9$, it is designed without residual blocks and consists of a shared convolutional feature extractor followed by two separate output heads for policy and value prediction. Shared feature extractor comprises three convolutional layers with output channels of 32, 64, and 128, respectively. Each layer uses a $3 \times 3$ kernel with padding 1 and applies ReLU activation. Policy head applies a $1 \times 1$ convolution to reduce channel size to 4, followed by a fully connected layer that outputs log-probabilities over actions using log softmax. value head applies a separate $1 \times 1$ convolution to reduce channels to 2, then passes through two fully connected layers to estimate the value of the current state. Final output is normalized to range $[-1, 1]$ using $\tanh$ activation. This architecture is equivalent to the AC structure described in the main text.

### C.2  Training setup

We evaluated a total of eight deep learning–based models, including non-distributional models (AC, QAC, DQN), their distributional counterparts (QR-AC, QR-QAC, QR-DQN), and two extended models that incorporate our proposed resource-aware MCTS framework (EQR-QAC, EQR-DQN).

Each model was instantiated with various combinations of hyperparameters ($N_{\text{mcts}}$, $N_\tau$), resulting in a total of 70 final agents used in the experiments.

All models were trained under the same fixed training procedure, following one of the methods proposed by Zheng et al. (2022). In each training iteration, the current best-performing agent was used to generate training data by playing 100 self-play games against itself. To encourage diversity in the training data, the temperature was set to 1 for the first 15 steps of each game and then fixed at 0 thereafter. Dirichlet noise was added to the root node of the MCTS tree during self-play to promote exploration.

Each training example was stored as a tuple (s, $\pi(a|s)$, r), where s denotes the game state, $\pi(a|s)$ the action distribution returned by MCTS, and $r$ the final game outcome. The policy network was trained using cross-entropy loss to match $\pi(a|s)$, while the value network was trained using mean squared error (MSE) loss to regress toward $r$. In each iteration, the network was trained for 10 epochs. After training, the updated network played 30 evaluation games against the current best model. If the new model achieved more wins, it replaced the current best network; otherwise, the original network remained unchanged. It is worth noting that the update criterion slightly differs from the one used in AlphaZero.

### C.3 Key differences in value networks

Here, we explain some differences in AlphaZero variants, especially of value networks. Note that the ones use policy network (AC-based AlphaZero) used the same policy network as AlphaZero explained in section C.1 of the Appendix.

#### C.3.1 Action value networks

The conventional Actor-Critic (AC) model used in AlphaZero and the QAC model employed in this study share a common structure in that both utilize a convolution-based feature extractor and are divided into a policy head and a value head. Additionally, the architecture of the policy head is identical in both models. However, the QAC model differs structurally in that it outputs action-value $Q(s, a)$ for all possible actions instead of a single state-value V(s).

In the QAC network, the value head is designed to produce $Q(s, a)$ for each action through a $1 \times 1$ convolution followed by two fully-connected layers. This structure is inspired by the dueling network architecture (Wang et al., 2016), enabling the simultaneous estimation of $Q(s, a)$ for all actions in a given state, thereby facilitating more precise value-based planning. As in AlphaZero, the Q-values for each action are trained using the final game outcomes obtained through self-play, and the learning is conducted by minimizing the MSE loss.

The DQN architecture used in this study differs from the original DQN proposed by Mnih et al. (2015) in both its network structure and training methodology. Our DQN consists of a single value network that outputs the action-values $Q(s, a)$ for all possible actions $a$, given a state $s$. By applying a softmax function to the resulting $Q(s, a)$ values, a probability distribution over actions can be implicitly derived, even without an explicitly defined policy distribution.

#### C.3.2 Models with quantile regression

Here, we provide details of how different models' value networks are different.

**QR-AC**    QR-AC architecture fundamentally follows the same neural network structure as the standard Actor-Critic (AC) model, and the policy head as well as the learning procedure remain identical. However, it differs in the value head, where instead of predicting a single scalar value V(s) for the state, it outputs $N_\tau$ quantile values, and their average is used as the final value estimate. The state value is computed as follows:

$$V(s) = \frac{1}{N_\tau} \sum_{i=1}^{N_\tau} \hat{V}_i(s)$$

The network consists of three convolutional layers followed by two fully-connected layers, with the final output representing the Q-values for all actions.

In contrast, the original DQN is trained using a temporal-difference (TD) loss, based on the reward R and the maximum estimated action-value at the next state s', i.e., $\max'_a Q(s', a')$. However, in our approach, similar to AlphaZero, the network is trained using the final game outcome instead of a discounted return, and the objective is to minimize the MSE loss.

**QR-QAC**    The QR-QAC architecture, like QAC, consists of a convolution-based feature extractor and separate policy and value heads, with the policy head sharing the same structure. However, the value head differs in that it outputs $N_\tau$ quantile values for each action $a$, instead of a single Q-value. The average of these quantile values is used as the final action-value $Q(s, a)$. The final Q-value is computed by averaging the predicted quantile values for each action as follows:

$$Q(s, a) = \frac{1}{N_\tau} \sum_{i=1}^{N_\tau} \hat{Z}_i(s, a)$$

Although the overall structure is similar to QAC, the learning methods differ. While QAC uses a MSE loss to regress a single Q-value toward the game outcome, QR-QAC directly optimizes each quantile output using a quantile regression-based Huber loss in order to learn the full distribution. This enables QR-QAC to estimate the uncertainty of action values more precisely.

**QR-DQN**    The QR-DQN architecture used in this study differs from the original QR-DQN proposed by Dabney et al. (2018) in both its network design and training methodology. Similar to our DQN structure, the QR-DQN in this study consists of a single value network that, given a state s, outputs $N_\tau$ quantile values for each possible action $a$. These values approximate the action-value distribution $Z(s, a)$, and their average is used as the expected action value:

$$Q(s, a) = \mathbb{E}[Z(s, a)] \approx \frac{1}{N_\tau} \sum_{i=1}^{N_\tau} \hat{Z}_i(s, a)$$

By applying the softmax function to the predicted expected values $\mathbb{E}[Z(s, a)]$, a probability distribution over actions can be implicitly derived without explicitly defining a policy. The network architecture consists of three convolutional layers followed by two fully-connected layers, identical to the structure of the DQN used in this study.

In contrast, the original QR-DQN constructs a target distribution by combining the reward R with the distribution Z(s', a') from the next state s', and learns by minimizing the Wasserstein distance between the predicted and target distributions. In this study, although we do not perform direct distributional matching with the next state's distribution, we still train the model to predict a full quantile distribution. Specifically, the predicted quantile values $\hat{Z}_i(s, a)$ are optimized using the quantile regression Huber loss with the self-play outcome as the target.

# D    ADDITIONAL RESULTS

## D.1    ELO RATING

Elo ratings were measured through a round-robin tournament among the trained agents, and the results are presented (Elo, 1978). Originally developed for two-player games such as chess, the Elo rating system quantifies the relative skill levels of players and dynamically updates scores based on match outcomes. In this experiment, each agent was initialized with an Elo rating of 1500, and a league-style tournament was conducted to enable a quantitative comparison of relative performance. Each agent played two matches against every other agent—once as the first player and once as the second. Elo scores were updated after each match using a fixed K-factor of 20. A total of 70 agents were evaluated, corresponding to the final-iteration models trained with various combinations of $N_{\mathrm{mcts}}$ and $N_\tau$ across eight different model types.

## D.2 ELO RATING OF ALL MODELS

Although we already provided a subset of results in the main text, here we present the full list of Elo ratings (Table 1). Results not shown in the main text were omitted because we included only one representative among several minor variants of each model. These minor variants differ only in hyperparameters, such as the value of $\epsilon$ in $\epsilon$-greedy.

In the main text, we considered non-linear $f(\cdot)$ in equation 2 as logarithmic, however, it might not true. Thus, we also tested different type of function $f(x) = x$ as well (Table 2). The results were consistent in terms of Elo rating, suggesting that the choice of non-linear function is not critical.

Table 1: Elo ratings of all models across different values of where $N_\tau$ discounts reward in logarithmic scale ($f(x) = \log_3 x$). Blue indicates the highest Elo rating among QAC models for a given $N_\tau$. Red indicates the highest Elo rating among DQN models for a given $N_\tau$. **Bold** indicates the highest overall Elo rating for a given $N_\tau$.

| Nuets | AC | QR-AC | QAC | QR-QAC | | | | EQR-QAC | DQN | QR-DQN | | | | EQR-DQN |
|---|---|---|---|---|---|---|---|---|---|---|---|---|---|---|
| | | | | $N_t = 3$ | 9 | 27 | 81 | | | $N_t = 3$ | 9 | 27 | 81 | |
| 2 | 1131.2 | 1160.6 | 1142.4 | 1141.1 | 1145.4 | 1141.9 | 1145.2 | **1165.2** | 1101.1 | 1108.2 | 1121.6 | 1113.7 | 1117.9 | 1119.3 |
| 10 | 1391.7 | 1428.2 | 1415.7 | 1418.8 | 1423.1 | 1449.4 | 1446.8 | **1459.1** | 1193.9 | 1247.3 | 1267.1 | 1259.2 | 1267.9 | 1271.5 |
| 50 | 1515.8 | 1555.4 | 1558.2 | 1573.9 | 1589.3 | 1595.5 | 1603.7 | **1652.9** | 1364.1 | 1377.1 | 1389.4 | 1399.4 | 1401.6 | 1401.6 |
| 100 | 1767.3 | 1791.9 | 1792.6 | 1809.8 | 1796.8 | 1809.5 | 1820.1 | **1837.4** | 1517.4 | 1554.3 | 1565.0 | 1552.4 | 1579.3 | 1579.3 |
| 400 | 1867.7 | 1893.7 | 1891.3 | 1907.2 | 1918.1 | 1912.6 | 1932.8 | **1941.1** | 1661.4 | 1718.3 | 1706.3 | 1720.5 | 1732.2 | 1747.2 |

Table 2: Elo ratings of all models across different values of where $N_\tau$ discounts reward in linear scale ($f(x) = x$). The color and bold rules are the same as in Table 1

| Nuets | AC | QR-AC | QAC | QR-QAC | | | | EQR-QAC | DQN | QR-DQN | | | | EQR-DQN |
|---|---|---|---|---|---|---|---|---|---|---|---|---|---|---|
| | | | | $N_t = 3$ | 9 | 27 | 81 | | | $N_t = 3$ | 9 | 27 | 81 | |
| 2 | 1131.2 | **1160.6** | 1142.4 | 1141.1 | 1145.4 | 1141.9 | 1145.2 | 1160.4 | 1101.1 | 1111.2 | 1121.6 | 1113.7 | 1117.9 | 1128.4 |
| 10 | 1391.7 | 1428.2 | 1415.7 | 1418.8 | 1423.1 | 1449.4 | 1446.8 | **1455.5** | 1193.9 | 1247.3 | 1267.1 | 1259.2 | 1267.9 | 1266.2 |
| 50 | 1515.8 | 1555.4 | 1558.2 | 1573.9 | 1589.3 | 1595.5 | 1603.7 | **1651.6** | 1364.1 | 1377.1 | 1389.4 | 1399.4 | 1401.6 | 1396.9 |
| 100 | 1767.3 | 1791.9 | 1792.6 | 1809.8 | 1796.8 | 1809.5 | 1820.1 | **1837.6** | 1517.4 | 1554.3 | 1535.2 | 1565 | 1552.4 | 1572.1 |
| 400 | 1867.7 | 1893.7 | 1891.3 | 1907.2 | 1918.1 | 1912.6 | 1932.8 | **1939.8** | 1661.4 | 1718.3 | 1706.3 | 1720.5 | 1732.2 | 1744.9 |

### D.3 OPTIMAL MOVE RATIOS OF ALL MODELS

We tested whether resource-aware models learn more efficiently from the early stages of training by comparing them with their conventional counterparts. For example, EQR-QAC was compared to QR-QAC and QAC, while EQR-DQN was compared to QR-DQN and DQN. The optimal move ratio is plotted as a function of training iterations (Figure 12, 13). We then analyzed the optimal move ratio as a function of move number, which is defined as the sequential order of actions within a game. This analysis focused on how well the models found the optimal move during the early in a game.

As shown, across $N_{\mathrm{MCTS}}$, across move numbers, it was consistent that the resource-aware model learns optimal earlier than any other models, both in terms of training iterations and in terms of move number in a game.

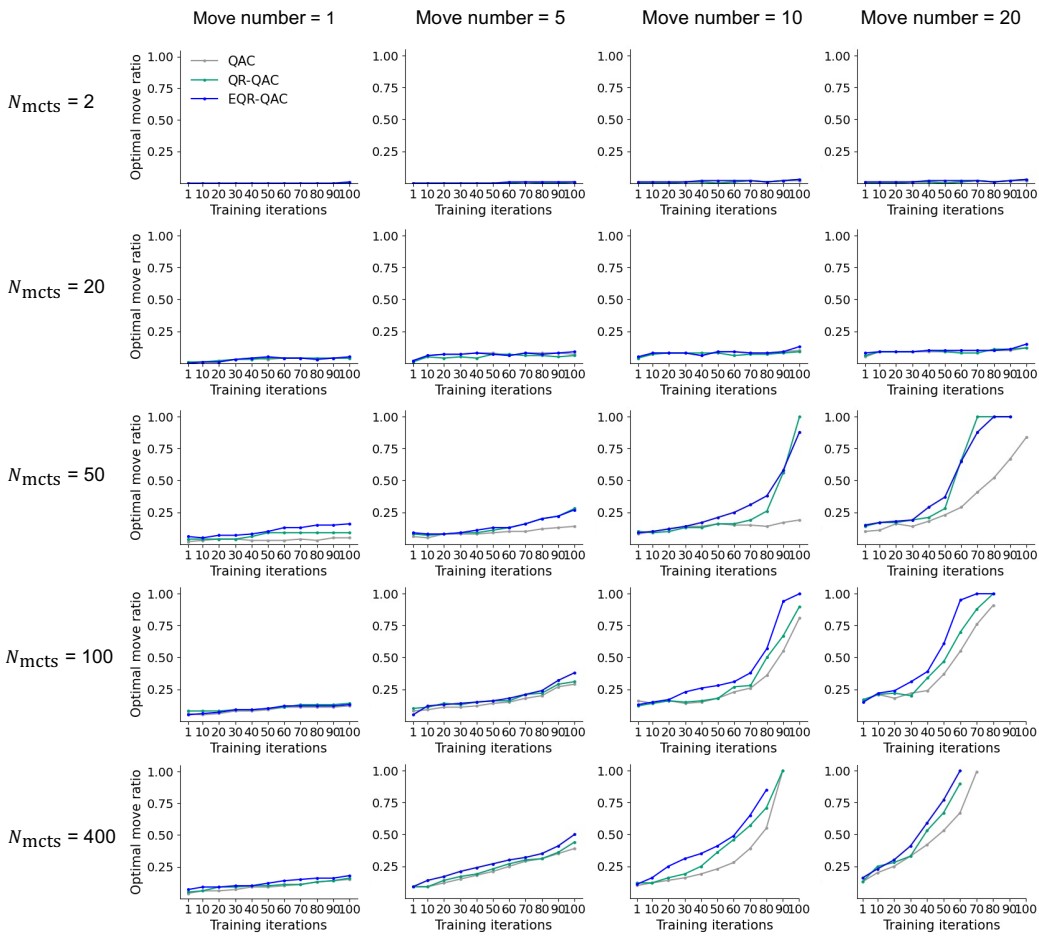

Figure 12: **full optimal move ratio (QAC)** The optimal move ratio of the QAC, QR-QAC, and EQR-QAC models was evaluated at various moves (1st, 5th, 10th, and 20th) and across five different $N_{\mathrm{MCTS}}$ values (2, 20, 50, 100, and 400). The optimal move is defined by a pure MCTS search with an infinite number of simulations ($N_{\mathrm{MCTS}} = \infty$), representing the fully searching model. The data is based on 100 inference iterations, with error bars indicating the corresponding error.

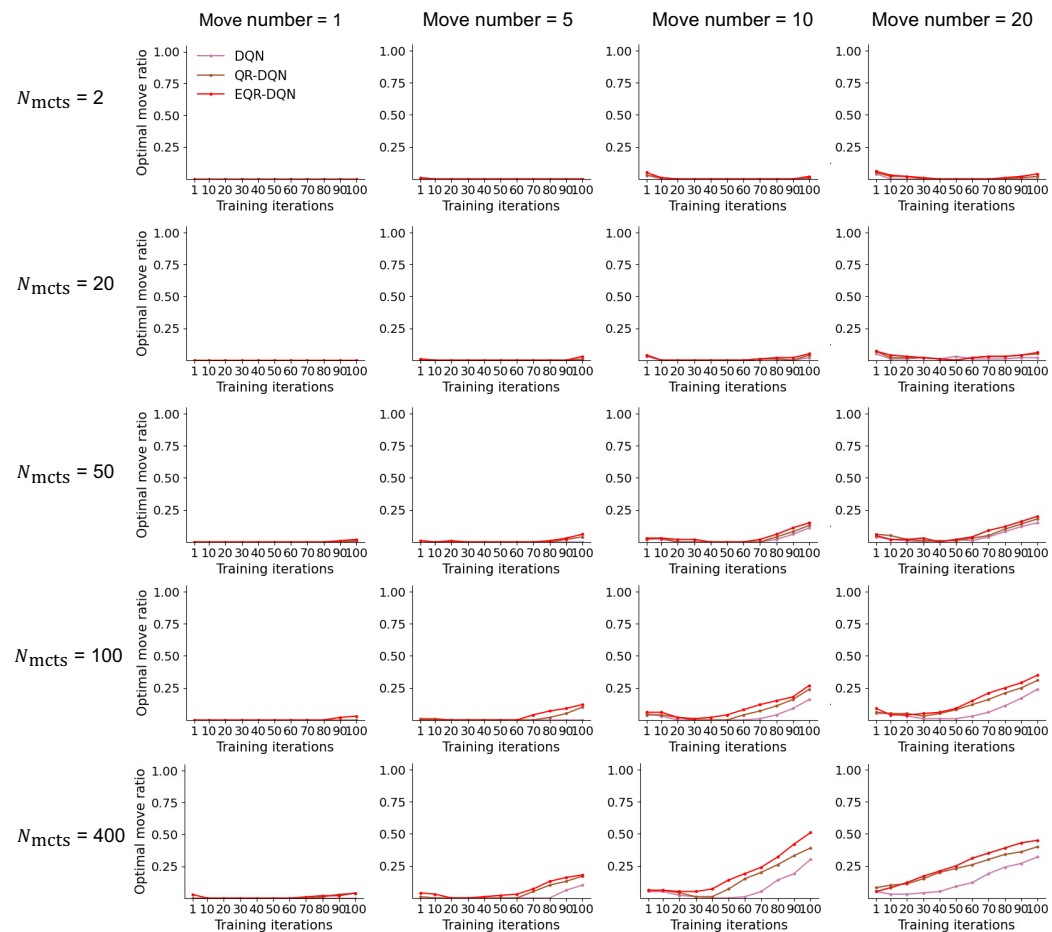

Figure 13: **full optimal move ratio (DQN)** Optimal move ratio of the DQN, QR-DQN, and EQR-DQN models at different move numbers. The results were obtained using the same method as the one used in Figure 12.

### D.4 RESOURCE ALLOCATION PROFILE AS TRAINING PROGRESSES

We examined how the model's resource allocation profile changes within a game evolves as training progresses (Figure 14). The planning depth increases across all move numbers as the training iteration grows, indicating that the model prioritizes planning depth as making more moves.

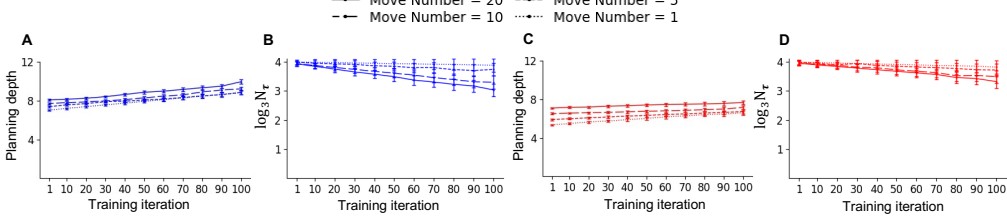

Figure 14: **Resource allocation profile with in a game as training progresses** (A, B) EQR-QAC's resource allocation profile, and (C, D) EQR-DQN's resource allocation profile. Different line style represents move number. Error bars represent standard deviations.

### D.5 HUMAN BEHAVIOR HEURISTICS OF ALL MODELS

We analyzed the difference between resource-aware models and their conventional counterparts to evaluate how closely the resource allocation behavior of our models resembles that of humans. For example, EQR-QAC was compared with QR-QAC and QAC, and EQR-DQN with QR-DQN and DQN. This comparison allowed us to confirm how well each model imitates human behavior (Figure 15 and 16).

To measure human-like behavior, we used four representative metrics: distance to the board center, distance to own pieces, number of threats made, and number of threats defended. (Kuperwajs et al., 2022; van Opheusden et al., 2023) Here, a "threat" is defined as a proactive attempt to build winning conditions, while "defense" is defined as an action that blocks an opponent's threats.

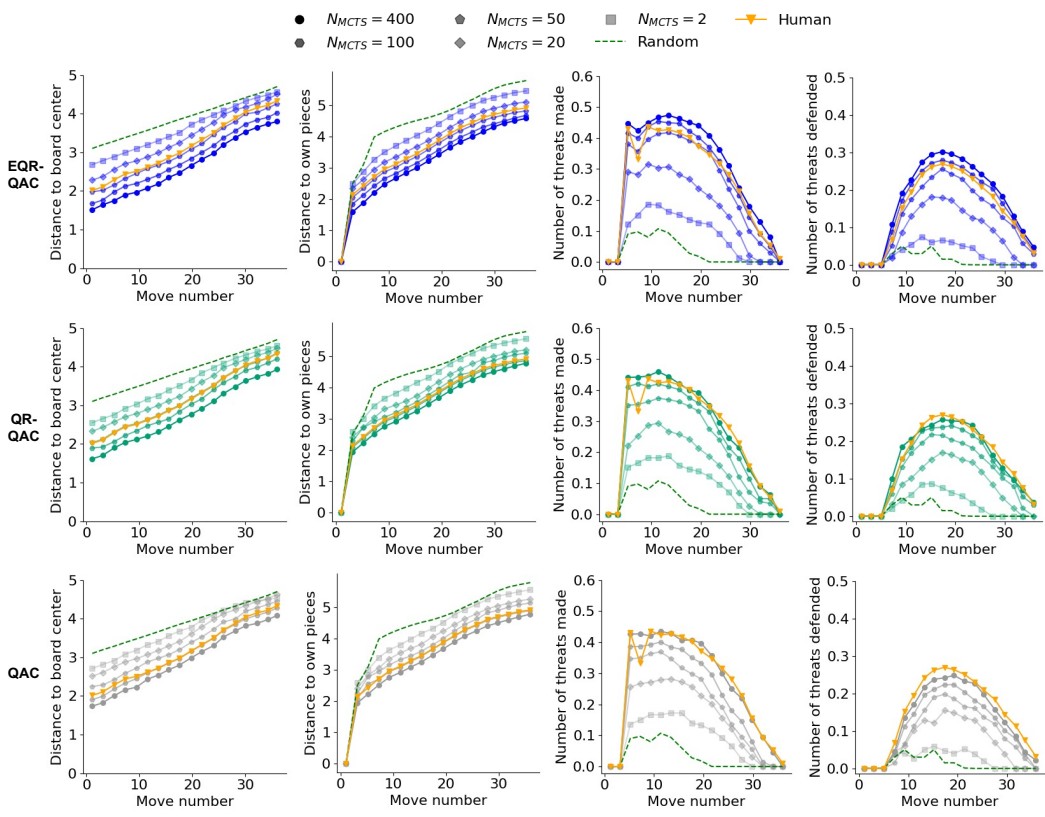

Figure 15: **Human behavior compared to QAC** We evaluated how closely QA-MCTS-based models (EQR-QAC) and their counterpart models (QR-QAC, QAC) resemble human play behavior under different $N_{\text{MCTS}}$ settings. Each model is compared against human behavioral data (red) and a random policy baseline (green), which selects actions uniformly at random.

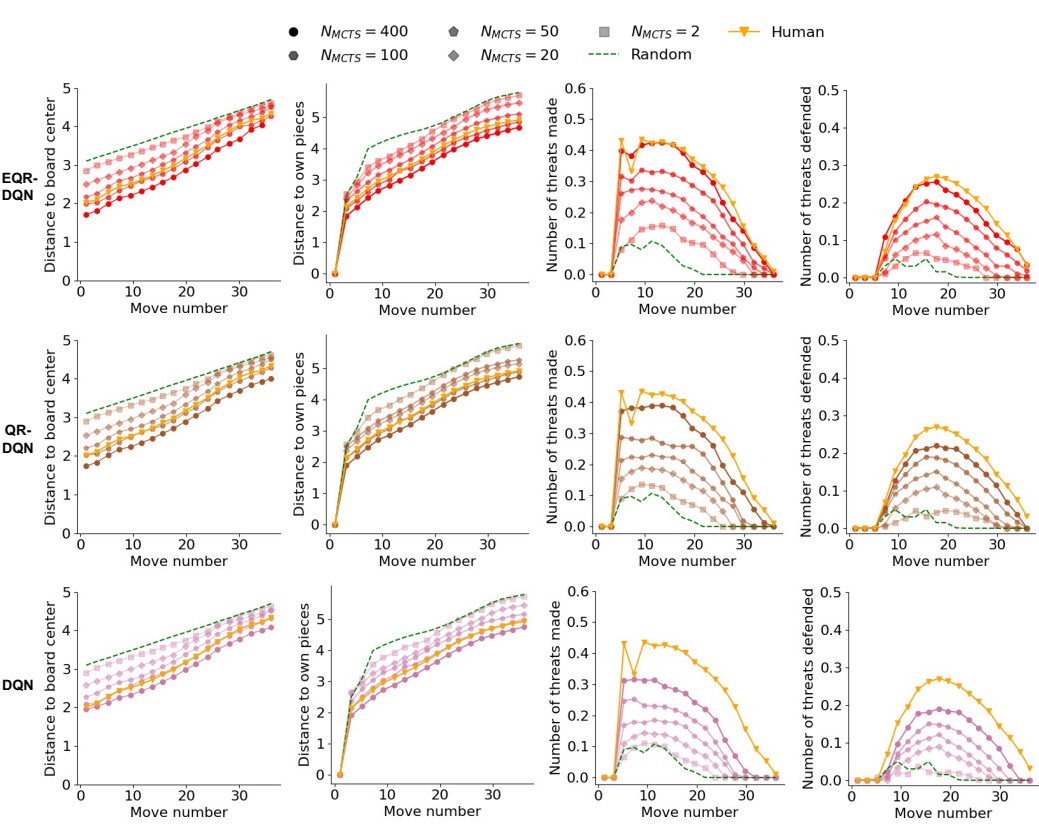

Figure 16: **Human behavior compared to DQN** We evaluated how closely QA-MCTS-based models (EQR-DQN) and their counterpart models (QR-DQN, DQN) resemble human play behavior unsder different $N_{\text{MCTS}}$ settings. All other experimental parameters and conditions were maintained as described in Figure 15.