# OpenReview forum: "Plan Deeply or Estimate Precisely?: A Resource-Aware AlphaZero with Dynamic Quantile Allocation"
_ICLR.cc/2026/Conference — ICLR 2026 Conference Withdrawn Submission_

### Official Review · Reviewer_NJLv · 2025-10-30

**Soundness:** 1
**Presentation:** 1
**Contribution:** 2
**Rating:** 2
**Confidence:** 3

**Summary:**

The paper proposes a resource-aware version of MCTS, based on AlphaZero, that models uncertainty in value estimates via quantiles, taking a page from the literature on distributional RL. The proposed technique adaptively modifies the number of quantiles to trade off between deeper planning and more accurate estimation. It was tested on a 4-by-9 four-in-a-row game.

**Strengths:**

The high-level motivation behind the paper is good and makes sense: Accounting for uncertainty in value estimates could be used to make MCTS-based search/planning more efficient.

**Weaknesses:**

The paper is very hard to understand in many places, and even unintelligible in some. There are severe difficulties with grammar, sentence structure, and exposition.

Some of the design and parameter choices are not motivated or explained.

Page 1:

AlphaZero does not employ policy gradients.

"the human, the most well-known intellectual, learns" -> "humans learn"

Page 2:

"distributional RL is the one that can" -> "distributional RL can"

"that predict a single" -> "that predicts a single"

"capturing both uncertainty either inherent or epistemic" -> "capturing uncertainty, either inherent or epistemic"

"let the AlphaZero aware of remaining" -> "inform AlphaZero of the"

"so that to dynamically allocate" -> "so that it dynamically allocates"

"tree searching depth" -> "tree search depth"

"—how uncertain" -> ", including how uncertain"

"for either coarse or fine-grained ones" -> "(either coarse or fine-grained)"

"Simply, the model is able to be more precise as the uncertainty (or variance error in Figure 1) reduces as using more quantiles to estimate value distribution." -> "Using more quantiles to estimate the value distribution allows the model to be more precise."

"Likewise, deeper planning reduce bias-error (as in Figure 1, makes the estimation of win or lose more accurate." -> "Deeper planning reduces bias-error, making the estimation of win or loss more accurate."

"how the proposing" -> "how the proposed"

"more earlier on" -> "earlier on"

"efficiently, and effectively" drop the comma.

"The proposed method tries to make less uncertain value estimation versus deeper
planning." This sentence is not intelligible.

"As number of quantiles increases" -> "As the number of qunatiles increases"

Page 3:

"that in human cognition, a" -> "that, in human cognition, a"

"game, each place" -> "game. Each player places"

"as the name suggests" drop this.

"since the size is proven to be where the optimal agent would not lose" Unclear what this means.

"allows more contrastive comparison of other AlphaZeros (or humans) to the proposed method" Unclear what this means.

"the intermediate state-space" -> "the intermediate state-space size"

"allows us to replace the state value estimation to action value estimation scheme, makes each node search in MCTS identical to a deeper search tree" This sentence fragment is unintelligible.

Page 4:

Figure 3: The description of MCTS in the caption of this figure is hard to follow; it appears to be incomplete and/or incorrect.

"the convergence characteristics of UCT is provided (Kocsis & Szepesvari, 2006)" -> "the convergence characteristics of UCT are provided in Kocsis & Szepesvari, 2006"

"equally distributed quantiles" Unclear what this means.

"which is proven to be the minimizer of Wasserstein distance" -> "which is adjusted to minimize Wasserstein distance"

Page 5:

"to share computational resource budget, for both number of" -> "to share the computational resource budget between the number of"

"in power law as" Ungrammatical.

"network, and training" -> "network and training"

"We provided proof sketch of convergence of the model as well (Appendix B)." -> "In Appendix B, we provide a proof sketch of convergence."

Page 6:

EQR-QAC is neither defined nor cited when it's first used.

"all models are in case of" -> "all models use"

"Elo rate" -> "Elo rating"

Page 7:

"meaning that the fully searching model" Ungrammatical.

"We tested if" -> "We tested whether"

"has compared to" -> "is compared to"

"which defined by" -> "which is defined by"

"in the same computational budgets (i.e., NMCTS) were applied" Ungrammatical.

"Appendix ??" Broken reference.

Page 8:

"Note that although resource-aware models has no NMCTS, but grouped together with the other models where their R is calculated with the same NMCTS (equation 3)." Grammatically unintelligible.

"This allow us" -> "This allows us"

"Regardless of training iterations, model has the same profile" Ungrammatical.

"NMCTS=400" missing spaces around = sign.

Page 9:

"used smaller" -> "uses a smaller"

"deeper regardless" -> "deeper, regardless"

**Questions:**

Questions

Page 3:

What do you mean by the following?

"AlphaZero may require an unrealistically enormous number of searches" To do what? AlphaZero works just fine in Go (against humans).

Page 5:

Why the colon argument in "π(s, :)"?

"predefined threshold θ = 0.1" Why this particular threshold? How did you choose it?

Page 6:

"In practice, we set f(Nτ) = log3 Nτ and g(NMCTS) = NMCTS (i.e., the identity function), which simplifies the implementation" How? This is not explained.

Do you have measurements of wall time and total memory usage for each experiment? These are very concrete measures of cognitive resources/costs, which could give a better idea of how efficient each method is in terms of resource rationality.

---

### Official Review · Reviewer_LzsG · 2025-11-01

**Soundness:** 2
**Presentation:** 2
**Contribution:** 2
**Rating:** 4
**Confidence:** 3

**Summary:**

This paper proposes a resource-aware variant of AlphaZero that dynamically allocates a fixed computational budget between deeper MCTS planning (bias reduction) and more precise value estimation through distributional RL using quantile regression (variance reduction). The algorithm incrementally increases the number of quantiles $N_\tau = 3^p$ until the action-value gap exceeds a threshold (\theta), otherwise spending remaining budget on tree depth. The total compute cost is modeled as
$R = f(N_\tau)\Delta_{\text{Quantile}} + g(N_{\text{MCTS}})\Delta_{\text{Depth}}$
with $f(N_\tau) = \log_3 N_\tau$ and $g(N_{\text{MCTS}}) = N_{\text{MCTS}}$.
Experiments on Four-in-a-Row demonstrate that the proposed EQR-QAC and EQR-DQN outperform fixed-resource baselines in Elo rating, exhibit faster convergence, adaptive budget allocation (depth vs. quantiles), and more human-like strategic behavior. Theoretical analysis provides quantile consistency and recalls UCT convergence; a full joint convergence proof remains informal.

**Strengths:**

1. The core idea of balancing bias–variance via dynamic quantile allocation under a resource budget is simple, elegant, and interpretable.
2. The compute-cost model is explicit and transparent, and results across architectures (AC, QAC, DQN) are consistent.
3. The behavioral analyses (adaptive search depending on opponent strength and human-like heuristics) are interesting and strengthen the cognitive motivation.
4. The quantile reuse strategy $N_\tau = 3^p$ effectively controls cost while maintaining precision.

**Weaknesses:**

1. Limited domain scope: All experiments are in Four-in-a-Row; no evidence of generalization to larger, continuous, or real-time environments.
2. Heuristic halting rule: The fixed action-gap threshold $\theta$ lacks systematic sensitivity analysis; the attempted “smoothed halting” is unstable.
3. Cost fairness concern: Only “new” quantiles are updated in EQR, possibly undercounting true compute vs. fixed-$N_\tau$ baselines; no wall-clock comparison.
4. Distribution collapse: Backups use the mean $\mathbb{E}[Z]$, losing potential advantages of full distributional propagation.
5. Theory remains partial: Joint convergence of the integrated algorithm is not proven, and no quantitative error–budget bound is provided.
6. Deviates from standard AlphaZero by using ε-greedy rather than PUCT, without analysis of the impact.

**Questions:**

1. Compute realism: How do the abstract cost functions $f,g$ map to real FLOPs/latency/energy? Are there wall-clock benchmarks?
2. Threshold sensitivity: How robust are results to different $\theta$ values? Could depth- or state-dependent thresholds stabilize halting?
3. Distributional backups: Why only back up expectations instead of full quantile distributions (e.g., via Wasserstein distance)?
4. PUCT vs. ε-greedy: What effect would using the standard PUCT rule have on search allocation and Elo performance?
5. Scalability and generality: Any plan to extend to continuous-control or time-critical domains with hard latency constraints?

---

### Official Review · Reviewer_s9q3 · 2025-11-02

**Soundness:** 4
**Presentation:** 3
**Contribution:** 4
**Rating:** 8
**Confidence:** 4

**Summary:**

This paper proposes a resource-aware MCTS combined with quantile regression RL.
At the tip node,
it dynamically increases the number of rollouts or terminates the rollout early,
based on the gap between the best and the second best SOMETHING.

The proposed change improves the sample efficiency.
This is empirically shown by ELO comparison and by computing the exact optimal policy,
which is made possible by the careful choice of the domain (four-in-a-row), which is clever.

I could not understand what this SOMETHING is, because the paper does not explain q and z in sufficient details.

**Strengths:**

-   Clear Motivations.
-   While the method is not explained enough, it is acceptable &#x2014; but fix it in the camera-ready.
-   Empirical evaluation is satisfactory.

**Weaknesses:**

So, I have two interpretations of SOMETHING.

Interpretation 1: gap between the best and the second best arm's estimated Q-values.
The estimates are rough when the quantile resolution is low (small p).
So, if the gap is below a threashold, it means the best and the second best arm are too close,
therefore, it increases the resolution for the quantile regression to obtain a clearer picture.

Interpretation 2: gap between the first and the second quantile in the CDF.
Isn' it just $1/ 3^p$ ?
If this is too small, the resolution increases &#x2026; which should make this gap even smaller.

Interpretation 1 is probably the right one because line line 264 mentions the action gap.
My confusion likely comes from the insufficient explanation of quantile regression in Fig 4 and the notations in Alg 1,2.
(What is q and z ? vector? scalar? )

minor:

-   line 264: (i.e.)
-   line 276: the notation is too liberal / informal. What does this E[z] mean?
-   Section 3,4: The text uses the acronyms (e.g. AC for actor critic) too liberally.
    You must expand them in the first appearance.
-   line 318: Is QAC defined elsewhere?

This paper is not compressed enough. Algorithm 1 and 2 can be compressed side-by-side
to explain all math symbols in them in detail.

Related work should mention various literature on metareasoning about
anytime/real-time decision making / heuristic search / planning and acting.
For example,

-   simple regret bandits that naturally models the selection of the incumbent best arm in an anytime fashion. I assume this paper uses PUCT instead for action selection.

    Sagers, Dominic, Mark HM Winands, and Dennis JNJ Soemers. "Anytime Sequential Halving in Monte-Carlo Tree Search." International Conference on Computers and Games. Cham: Springer Nature Switzerland, 2024.

-   deciding how long to think (plan) before acting.

    Bhatia, Abhinav, et al. "Tuning the hyperparameters of anytime planning: A metareasoning approach with deep reinforcement learning." Proceedings of the International Conference on Automated Planning and Scheduling. Vol. 32. 2022.

**Questions:**

Please confirm my interpretation above.

Explain q and z's representation.

Are you sure that there are no existing metareasoning approaches to MCTS or other search algorithms?
(I doubt it. There should be some, and you should at least mention it. Doesnt mean additional experiments are necessary, though.)

---

### Official Review · Reviewer_jDiC · 2025-11-03

**Soundness:** 3
**Presentation:** 2
**Contribution:** 2
**Rating:** 4
**Confidence:** 4

**Summary:**

The paper aims to extend AlphaZero (AZ) with a distributional value representation using quantile regression (QR). In analogy to human decision making, the authors aim for a ressource-constraint AZ, which dynamically decides to either plan deeper or increase the agent's certainty in the value estimates. The value estimates can be estimated by a variety of expected and distributed baselines (AC,QAC,DQN and their QR-versions), and the proposed version (EQR-QAC and EQR-DQN) either increase the number of quantiles or plans deeper, depending on the gap between expected value estimates of the two best actions. The methods and all baselines are evaluated on a 4x9 four-in-a-row game and show moderate ELO increase. The authors also evaluate the optimal move ratio and the empirical planing depth and number of evaluated quantiles. To link the approach to human cognition, the authors measure 4 human heuristics from literature.

**Strengths:**

I like the idea a lot, and a quick search did not reveal prior papers that extend AZ with distributional value estimators, which shows the novelty of the approach. Testing on a simplified game is a good choice, and the link to human planning behavior is (at least conceptually) quite interesting.

**Weaknesses:**

The paper contains a number of minor issues (see below), but my main recommendation to reject it is based on some crucial flaws (or misunderstandings on my part):
1. Algorithm 2 contains a call to "Quantile regression" with a variable number of quantiles, but I fail to see what is happening here. Conceptually this should call the neural network with the current node-state and action as input to estimate the quantiles of the return distribution. However, this network usually has number-of-quantiles outputs, so why would we constraint the number of used outputs? And why would this be called "Quantile regression", which indicates a training of the network parameters as in Equation 1?
2. Algorithm 2 uses only the expectation of the quantiles ($q' \leftarrow \mathbb E(z)$), which are then returned and back-propagated. This misses the point for me: there are two ways to become more "certain" about the return distribution, either estimate it in more detail (more quantiles), or plan deeper (more depth) to see what the actual outcomes were. But Algorithm 2 never propagates distributions in the decision tree, which would allow to compare the two. Instead it only uses distributional network to estimate the expectation and treats that expectation as if it would have come from a simple expected value function.
3. The approach is motivated (Figure 1) as a trade-off between precision (error variance) and depth (bias error). However, as far as I can see, the error variance is never evaluated/used and the "ressource-aware" criterion thus does not make sense to me. Algorithm 2 increases the number of quantiles until the (predicted) expected returns of the best two actions are farther apart than some threshold. But that has nothing to do with any certainty that one action is better than another! What the authors *should* (or at least could) have done is to compare the overlap between the return distributions of the two best actions. If they have no overlap with few quantiles (at least if the few quantiles overestimate the true variance), then adding more will not change which action is better. There can still be situations where two best actions cannot be confidently preferred over each other (if the overlap is large even with many quantiles), but this case *should* be handled in the "NodeSelect" function (which is never defined and I assume it is the default from AZ). In the current form I cannot see why Algorithm 2 should make a sensible trade-off.
4. It remains unclear *which resource* is constrained here. Above I argue why I do not see the computational advantage of using less than all quantiles that have been learned during training, but even in the evaluation this seems unclear. Empirically the E-variants are searching somewhat deeper, which could already explain the improved performance. This is because their baselines all seem to use a fixed resource $N_{mcts}$, or do I misinterpret this? For a *fair comparison*, the baselines (e.g. AC/QAC/DQN) would be given the same "resources", whatever they are. If the authors want to argue that fewer quantiles require less resources (whyever that may be) then DQN should only consume the equivalent of one quantile in resources. This should allow the expected value methods to plan significantly deeper than the proposed methods, and thus perform much better than they do. Unless this *is* how the authors implented it, then they must add those baselines to Figure 7. In general I am missing a baseline of EQR-* where everything else remains the same (especially total available resources), but $N_\tau=3$ is fixed (while loop breaks after the first iteration).
5. The comparison with human characteristics, while a nice contribution, do not support the conclusion "the resource-aware AlphaZero is the best model that replicates human heuristics" and that "All the others [..] does not capture human heuristics". Instead the curves seem to be mainly sensitive to final performance. The better an agent does, the more it resembles (or surpasses) the human in the tested heuristics. The fact that DQN does not reach the human characteristics could be explained by its poorer performance. What is missing is a comparison with the human performance to evaluate this alternative hypothesis.

I am happy to increase my rating if these concerns (especially 1-4) can be alleviated, but I doubt this can be done in a simple revision.

**Detailed comments**
- l.38: It is not clear what "uncertain environments" mean here, aleatoric (irreducibly stochastic transitions) or epistemic (reduces with more training data for value function) or planning (reduces with more planning steps).
- l.43: AZ does not use "policy gradient algorithms", but a cross-entropy loss to the search probabilities, which are proportional to (sometimes a power of) the visitation counts at the root. This seems to be very different from the $\epsilon$-greedy approach you mention in ALgorithm 1. Where did you get this idea from (maybe you just need to cite it)?
- l.58: The discussion on "abstraction" is somewhat unclear, as the one-to-one correspondence in AZ are the neural network generalizations.
- l.66: The assertion that quantile regression "capturing both uncertainty either inherent or epistemic" is misleading. As far as I know, QR captures aleatoric uncertainty only. Epistemic uncertainty, like how much one can rely on the network predictions based on how similar the state is to those seen during training, is *not* estimated.
- Fig.3: the two evaluation subfigures are confusing: "select" indicates the right node is expanded, but the the left node is evaluated. It is also unclear what the arrows to the policies mean.
- l.184: AZ does not use UCT, but P-UCT [1], which explains why it needs a policy $\pi$ in the first place.
- l.192: Please check the MCTS/AZ literature. The statement "treating those nodes as 'promising' under a utility function" is inaccurate, as both UCT and PUCT are not merely arbitrary "utilities" actually upper bounds on the true value under certain assumptions, which is why one can prove convergence.
- sec.2.4: The description of QR-DQN is too superficial to understand here (like where do the $y_j$ come from), and a description/citation of QR-QAC is missing.
- sec.3.3: It took me a while to understand what you mean with "backbones". My interpretation is that you estimate the (distributional) value functions with "backbone algorithm" X and then use that in AZ (which is *not* how vanilla AZ does it). If I am right, please just say this at the beginning. The choice of $f(N_\tau)$ also seemed weird at fist, but it depends on which "resource" you actually constrain (see above).
- fig.5: I know ELO computation is expensive, but without standard deviations it is hard to gauge whether any advantage between algorithms is actually significant. Differences seem to be small in comparison to deeper planning.
- fig.6: This is a much better argument than Fig.3 (because it contains standard deviation bars), but the optimal move ratio at specific time steps is highly correlated with general performance and search depth, because well performing agents win earlier and thus are closer to terminal states where they do not need to estimate the value anymore. It is also not clear what a "training iteration" is here. A game? An epoch over some replay buffer?
- l.359: missing Appendix
- fig. 7,8,10: I cannot see which standard deviation bars overlap which here, which makes the whole plot fairly inconclusive.

**References**

[1] Rosin, C. D. Multi-armed bandits with episode context. Ann. Math. Artif. Intell., 61(3):203–230, 2011. URL https://link.springer.com/article/10.1007/s10472-011-9258-6.

**Questions:**

See above.

---

### Note · Authors · 2025-11-17

**Comment:**

Sorry for the reviewers. While checking the details of our implementation, we found a serious error. We are grateful for the reviewers comments. We are planning to resolve the issue and will submit next time!

**Withdrawal Confirmation:**

I have read and agree with the venue's withdrawal policy on behalf of myself and my co-authors.